# On the Effect of Standard Deviation of Cationic Radii on the Transition Temperature in Fluorite-Structured Entropy-Stabilized Oxides (F-ESO)

**DOI:** 10.3390/ma16062219

**Published:** 2023-03-10

**Authors:** Luca Spiridigliozzi, Mauro Bortolotti, Gianfranco Dell’Agli

**Affiliations:** 1Department of Civil and Mechanical Engineering, University of Cassino and Southern Lazio, Via G. Di Biasio 43, 03043 Cassino, Italy; 2Department of Industrial Engineering, University of Trento, Via Sommarive 9, 38123 Trento, Italy; 3INSTM—National Interuniversity Consortium of Materials Science and Technology, Via G. Giusti 9, 50121 Florence, Italy

**Keywords:** entropy-stabilized oxides, fluorite-structured ceramics, high-temperature XRD diffraction, entropy-driven single-phase transition

## Abstract

It is confirmed that Fluorite-structured Entropy-Stabilized Oxides (F-ESO) can be obtained with multicomponent (5) equimolar systems based on cerium, zirconium, and other rare earth elements, selected according to the predictor already proposed by the authors. Indeed, in the present study, three different samples owning a standard deviation (SD in the following) of their cationic radii greater than the threshold value (i.e., SD > 0.095 with cationic radii measured in Å) needed to ensure the formation of the single-phase fluorite structure, were prepared via co-precipitation method. After a calcination step at 1500 °C for 1 h, the entropy-driven transition from multiple phases to single-phase fluorite-like structure has been actually confirmed. Thus, with the aim of defining the temperature at which such entropy-driven transition occurred, and identifying possible relation between such temperature and the actual value of SD, the phase evolution of all the prepared samples as a function of temperature (ranging from 800 °C to 1300 °C) was analyzed by in situ High Temperature X-ray Diffraction. An apparent inverse correlation between the standard deviation and the entropy-driven transition temperature has been identified, i.e., the higher the former, the lower the latter. These results, based on the conducted basic structural analysis, provide further support to the SD-based empirical predictor developed by the authors, suggesting that high values of SD could bring additional contribution to the overall entropy of the system, other than the configurational one. Thus, this SD-driven entropy contribution directly increases with the increasing of the standard deviation of the cationic radii of a given F-ESO.

## 1. Introduction

The pioneering work of Rost et al. [1] demonstrated for the first time that an equimolar multicomponent oxide (Mg_0.2_Ni_0.2_Co_0.2_Cu_0.2_Zn_0.2_O) could be stabilized by entropy into a single-phase rock-salt structure. Rost’s study paved the way for a huge amount of worldwide research devoted to both ESOs, i.e., multicomponent oxides characterized by the reversibility between low-temperature multiple phases and high-temperature single phase, and High-Entropy Oxides (HEOs), i.e., oxides owning a high configurational entropy (>1.5 R, where R is the gas constant), but not the reversibility of the multiple phases to single-phase transition. Reversibility requires high configurational entropy and positive enthalpy of formation. Therefore, ESOs and HEOs are different even though they are often used interchangeably. The compositional flexibility of such compounds, i.e., the ability to maintain phase purity despite variation in composition, provides opportunities for fine-tuning their properties [2].

Several aspects of these new materials were thoroughly investigated during these last few years. First, ESOs and HEOs owning other crystal structures than rock-salt were discovered and synthesized, such as fluorite [3,4], perovskite [5,6], pyrochlore [7], magnetoplumbite [8], garnet [9], and spinel [10]; a detailed review about the compositions for different families of HEOs was authored by Akrami et al. [11]. Then, several papers reported very promising, and often unexpected, technological properties for both ESOs and HEOs, especially in the field of energy storage systems [12,13,14], thermal barrier coatings [15,16], multipurpose catalysts [17,18,19], and magnetic systems [20]. Later on, both alternative synthesis methods, such as wet-chemical methods [4,21], combustion synthesis [22], or modified sol-gel routes [23], innovative thermal consolidation routes, such as Flash-Sintering [24], Ultra High Sintering [25], and Spark Plasma Sintering [26], were proposed in the literature to obtain high-entropy materials.

Simultaneous to these great efforts aimed at discovering novel ESO/HEO compounds, potential practical applications, and engineered processing methods, designing specific compositions knowing a priori (i.e., without testing it experimentally) if they are able to form an ESO or HEO, and with which crystal structure, is becoming increasingly important. In other words, a particularly interesting point consists of the preliminary identification of systems able to be entropy-stabilized in a certain single-phase, possibly also knowing a priori the temperature at which the multiple phases to single-phase transition occurs. Indeed, from a practical point of view, a certain ESO system is technologically interesting if its transition temperature is not excessively high.

In the case of fluorite-structured ESOs based on cerium, zirconium, and rare earth elements, the authors were able to propose and discuss a very simple model to predict their formation [27,28]. In particular, the authors showed that there exists a threshold value of the standard deviation (SD) of the involved cationic radii beyond that a fluorite-structured single-phase compound is obtained, as long as the fluorite structure could be “written” at a such temperature on the base of the cluster-plus-glue atom model [29,30]. SD is calculated by the following equation:(1)$$\mathrm{SD}=\sqrt{\frac{\sum_{i=1}^{5}{\left(r_{i}-\bar{r}\right)}^{2}}{\left(5-1\right)}}$$
where the sum is extended to the five cations of the sample, and $\bar{r}$ is the average cationic radius.

In these previous studies, the tested temperature was 1500 °C, but the same authors have also shown that for the Ce_0.2_Zr_0.2_La_0.2_Gd_0.2_Y_0.2_O_1.7_ system, the transition temperature lies in the range of 1100 °C to 1200 °C [4].

Therefore, the identification of the transition temperature for each possible equimolar F-ESO remains a still open question. In other words, is it possible to identify the exact entropy-driven transition temperature for an ESO or, at least, to identify what parameters influence it?

It is known that the temperature of transition is related to competition between entropy and enthalpy according to the following equation:(2)$$\Delta G=\Delta H-T\Delta S=0\Rightarrow T=\frac{\Delta H}{\Delta S}$$
while through quenching, the high-temperature stable single-phase may become metastable at room temperature [31]. 

Starting from our previous results [27,28], we undertook the present study based on in situ high-temperature X-rays Diffraction aiming to unravel possible relationships between the specific value of SD of a certain F-ESO and its entropy-driven transition temperature. In particular, the in situ high-temperature X-ray diffraction has been used to avoid possible practical problems deriving from the eventually unpredicted effect on the cooling rate.

Therefore, the main purpose of this study lies in the fact that even though our empirical predictor indicates a SD threshold value for F-ESOs, it is reasonable to hypothesize that varying the value of SD, in any case beyond that threshold, some effects on the transition multiple phases ↔ fluorite-like single-phase could appear. To this end, three F-ESO systems owning different SD (all higher than the threshold value of SD = 0.095), were synthesized via co-precipitation and characterized through in situ high-temperature X Rays Diffraction and thermal analysis; the composition of the three systems was selected to have a first sample with SD only slightly higher than 0.095, a second sample with SD well higher than 0.095, and a third sample with SD intermediate to the last two ones. The obtained results, based on the conducted basic structural analysis, suggest that the F-ESO entropy-driven transition temperature may effectively change according to different SD and a trend can be highlighted.

## 2. Experimental

Three different samples, whose composition was formed by five cations (chosen among Ce, Zr, and different Rare Earths) in an equimolar ratio, owning different values of SD as explained above (near, far, and far more than the threshold value of 0.095, i.e., 0.0976, 0.117, and 0.128, respectively) were prepared by a coprecipitation route. Indeed, to ensure a very good mixing at the atomic level, i.e., an essential prerequisite in such a complex system, and to avoid very-long ball milling treatments, wet-chemical methods as coprecipitation, hydrothermal treatment, and combustion synthesis are frequently adopted for ceria-based and zirconia-based systems [32,33,34]. The used raw materials were hydrated nitrates for the Rare Earths, including cerium (all Reagent Grade with >99% purity and supplied by Sigma-Aldridge), while zirconium oxynitrate was the used precursor for zirconium (ZrO(NO_3_)_2_—Carlo Erba, Milan, Italy, with purity > 98%). As precipitating agent, an ammonia solution 30 wt% (Reagent Grade, Carlo Erba, Milan, Italy) diluted at about 4 M was used. Briefly, the ammonia solution was quickly added to the solution of selected cations with the instantaneous formation of the co-precipitate; after an aging of 15 min, the co-precipitate was recovered by filtration, washed several times with demineralized water, and finally dried overnight. All the details of the synthesis can be found in a previous work [27].

Table 1 reports the chemical formulas and the labels of the synthesized samples, along with the cationic radii of the cations present in each sample, their related SD, and the average radius. SDs have been calculated by considering the exact values of the cationic radii in the VIII coordination from [35].

The as-synthesized samples were previously calcined at 800 °C before the in situ high temperature X-ray to allow the evolution of all the gaseous species possibly impairing the in situ diffraction run. In addition, a batch of each sample was treated at 1500 °C for 1 h in a conventional furnace, followed by air quenching at 1250 °C, to ascertain that the expected fluorite-like single phase was formed.

In situ high temperature experiments were performed by means of an Italstructures IPD3000 diffractometer equipped with a monochromatic Cu-kα source, a 1024 channels Dectris Mythen 1K detector, and an Anton Paar HTK16N furnace operating in an air atmosphere.

For each sample, a powder/ethanol slurry was deposited on the chamber heating element (1 mm thick platinum strip), then let dry in air, forming a powder film of about 100 μm thickness. Diffraction data was then collected in reflection geometry (with a fixed 5° incidence angle) on a 20–90° 2θ interval (0.02° step resolution) for a total of 30 min of acquisition time per pattern. In situ experimental data for each sample was collected in air atmosphere. After a first room temperature reference data acquisition, the sample was brought at 800 °C with a 20 °C/min heating rate; further powder diffraction patterns were then collected at 800 °C, 900 °C, 1000 °C, 1100 °C, 1200 °C, and 1300 °C, respectively, with a 10 °C/min heating rate and after a 60 min resting interval at each temperature step. The diffraction patterns collected by in situ high-temperature experiments were analyzed through the software High Score Plus (Panalytical, Almelo, The Netherlands) to extract the diffraction peaks data. Bragg’s angle of all these peaks was used to calculate the lattice parameter of the fluorite-like phases by means of the least-square procedure implemented in the software Unit Cell [36] referring to ICDD card n. 34-394. Crystal size was calculated by Scherrer’s equation applied on the strongest peak of fluorite, i.e., (111) reflection. Diffraction patterns of samples calcined at 1500 °C were analyzed by Rietveld refinement by using the ReX powder diffraction software [37]; the modeling was performed starting from the fluorite reference structure by adopting arbitrary reflection intensities, and refining the cubic lattice and size and strain broadening parameters in the isotropic approximation.

The thermal behavior (Thermogravimetric Analysis, TG, and Differential Thermal Analysis, DTA) of the as-prepared samples were investigated through a STAR analyzer (Mettler-Toledo, Columbus, OH, USA) in air; with a heating rate of 10 °C/min up to 1300 °C and using α-Al_2_O_3_ as a reference.

## 3. Results and Discussion

As reported in Table 1, all the analyzed samples own a SD of the involved cationic radii greater than the threshold value reported in our previous work [27], approximately equal to 0.095.

Therefore, after calcinating the as-precipitated samples (being all very poorly crystallized, as shown in Appendix A, where a representative XRD pattern of Int_SD sample is shown) at a sufficiently high temperature (i.e., higher than the entropy-driven transition temperature), the expected single-phase is a fluorite-like phase (and not a bixbyite-like or mixture fluorite/bixbyite, as in the case of SD lower than the threshold value). As a confirmation of that, Figure 1 shows the diffraction patterns of the three samples calcined at 1500 °C for 1 h, being all single-phase fluorite-like systems, even though an evident shift of Bragg’s angles is displayed, indicating different values in lattice parameters. In Table 2, the lattice parameter and the crystal size of the sample deriving from Rietveld refinements are reported. The plots of the Rietveld refinements are shown in Appendix A, revealing very good fittings. There is a perfect agreement between the average ionic radius (see Table 1) and the lattice parameter according to the following relation:(3)$$a_{Low\_SD}<a_{Int\_SD}<a_{High\_SD}$$

The samples calcined at 1500 °C are all well crystallized, even though some differences appear in the values of crystal size. Noticing that in Table 2 the order of crystal size is opposite to the order of lattice parameters, one can argue that the greater the average cationic radius, the lower the crystal growth; this is a reasonable result, as the mobility of larger cations is very likely lower than the mobility of smaller cations.

To determine the right calcination temperature for the three samples, the thermal behavior of the three prepared samples was preliminarily analyzed.

The thermal and thermogravimetric analysis of the as-prepared samples, showed in Figure 2, highlights many similarities, but also several differences.

First, despite the presence of a relevant amount of amorphous phase in the as-synthesized products, as revealed in Appendix A, in all three DTA curves, there is no trace of exothermic crystallization peaks. It has been reported [38] that precipitation with ammonia of ceria or doped-ceria systems leads to partially crystallized products owning fluorite-like structure, which undergo a crystallization event clearly visible in the DTA curve. In the absence of specific analysis to reveal the crystallization path (well outside the purpose of the present work) in the studied systems, a reasonable deduction is that, due to their great chemical complexity, the crystallization gradually proceeds during the heating over a wide temperature range and, instead of leading to a bixbyite-like phase (the typical crystal structure for RE_2_O_3_ compounds, where RE stands for a generic trivalent Rare Earth), the preferred destination of such a process is the dissolution of rare-earth oxides (and zirconium dioxide) into the fluorite-like phase. As a result, after that process, all three systems are formed mainly by the fluorite-like phase along with a minoritarian one, very likely bixbyite, as explained below in the discussion of Figure 3, Figure 4 and Figure 5.

The thermogravimetric plots show similar trends for sample High_SD and Low_SD, being both characterized by a global weight loss slightly greater than 20%, distributed in three main thermal events, corresponding to three endothermic events in the related DTA curves, whose peaks are centered at about 200 °C, 500 °C, and 700 °C (marked in Figure 2 with α, β, and γ, respectively). The first thermal event (α) is related to the evolution of adsorbed water on the powders [4], whereas the other thermal events (β and γ) are related to the thermal decomposition of hydroxides/hydrated oxides formed upon co-precipitation [4]. Anyway, in both cases, the thermal decomposition ends at about 800 °C. Apparently, the thermogravimetric plot of sample Int_SD is different, being characterized by a total weight loss close to 30%; additionally, by carefully analyzing it, one can note that Int_SD weight loss up to 200 °C is well greater than the corresponding one in the same temperature range for High_SD and Low_SD (around 4% for Int_SD and around 10% for both High_SD and Low-SD). In other words, the difference of total weight loss in the sample Int_SD is partly due to a larger amount of physically adsorbed water, very likely due to random reasons, such as external moisture conditions (the three samples were prepared in different days), and not to substantial differences in samples’ thermal behaviors. As a confirmation of that, the DTA plot of sample Int_SD is rather similar to the other ones, even though its thermograph exhibits a drift likely masking the event γ. However, there is a noteworthy difference between the DTA curve of Low_SD and the DTA curves of Int_SD and High_SD: the absence in the former of a further endothermic peak (indicated with “*” in Figure 2) appearing at high temperature for Int_SD and High_SD at about 1250 °C and 1150 °C, respectively. As these endothermic peaks are not associated with any decomposition event (no weight loss is associated with them), they could be related to a phase transformation and, in particular, to the entropy-driven transition from multiple phases to the fluorite-like single-phase. As for sample Low_SD, the absence of an analogous peak could be explained with the hypothesis that the entropy-driven transformation occurs outside the range of temperature investigated in the DTA-TG runs (i.e., the transition temperature should lie above 1300 °C). In conclusion, the findings deduced from the thermal analyses are the following ones: the decomposition events ended for all samples at around 800 °C, and there is indirect evidence of the entropy-driven transition from multiple phases to fluorite-like occurring at temperatures lower than 1300 °C for samples High_SD and Int_SD, consequently suggesting that there exists a correlation between SD and the typical ESO entropy-driven transition.

Thus, after a calcination step at 800 °C for 1 h in a conventional furnace, all the samples were analyzed in HTK16, collecting their diffraction patterns at temperatures ranging from 800 °C to 1300 °C after a dwell time of 1 h. Such diffraction patterns are displayed in Figure 3, Figure 4 and Figure 5 for the samples Low_SD, Int_SD, and High_SD, respectively. It should be noticed that in several diffraction patterns some additional signals, not belonging to crystallographic phases of interest for the current study, can be recognized, likely originating from the platinum sample holder or other mechanical parts of the furnace. In particular, the peak located at around 79° in the diffractogram collected on sample Low_SD at 1300 °C is likely due to the (311) reflection of Pt cubic lattice, whereas the signals present at about 45° in all H.T. data collected on sample Int_SD can be ascribed to Pt (200) reflection. Additionally, peaks around 65° found in patterns 800 °C, 900 °C, and 1000 °C of sample Int_SD and 1300 °C of High_SD are possibly due to the Pt (220) reflection, while signals located at about 78° in all High_SD H.T. data are of unknown origin, and could be due to some generic, unshielded scattering from the furnace steel frame. Even though some authors prefer to remove “a posteriori” these spurious signals from the diffraction patterns, generally speaking, we rather preferred to leave them. Anyway, these spurious signals are properly marked in all the following Figures.

That said, at first glance, all the diffraction patterns in Figure 3, Figure 4 and Figure 5 appear almost identical, being all characterized by a main (or the only) fluorite phase. However, non-negligible differences among the patterns displayed in Figure 3, Figure 4 and Figure 5 are associated with a minoritarian bixbyite-like phase (if present) whose peaks are indicated with “*”; the appearance and disappearance of such weak diffraction peaks, associated with bixbyite phase, as a function of the calcination temperature is characteristic of the fluorite-structured entropy stabilized oxides [4,27]. Indeed, by increasing the collecting temperature, such weak peaks tend to disappear in the diffraction patterns. Moreover, at 1300 °C both High_SD (Figure 5) and Int_SD (Figure 4) are fully fluorite-like single phase, as in these diffraction patterns are recognizable only peaks associated with fluorite, whose reflections are marked with “F”, with the exclusion of the artifacts (highlighted with a “§”) close to 70° 2θ for the sample High_SD and close to 45° 2θ for the sample Int_SD. On the contrary, at 1300 °C Low_SD is still multiphase due to the presence of weak additional peaks (see peaks indicated with “*” in Figure 3), being not artifacts of the instrument. Moreover, by analyzing all the different diffraction patterns at decreasing temperatures, it can be noted that sample High_SD exhibits additional phase (see peaks indicated with “*” in Figure 5), at 1000 °C, whilst sample Int_SD exhibits additional phaseat 1100 °C (the related additional peaks are marked with “*” in Figure 4). These findings are in very good agreement with the results obtained from DTA thermograms, as the DTA runs are dynamic analyses so that the recorded temperatures for the different thermal events are pushed forward compared to the static runs of the HTK16. In other words, the endothermic peaks marked with “*” in Figure 2 for samples High_SD and Int_SD could be assigned to the ESO entropy-driven transition from a multiphase system to a single-phase system. Moreover, from the HTK16 results, it seems evident that the transition temperature is in inverse correlation with the SD of a certain ESO system. Indeed, for high SD systems (i.e., sample High_SD) the transition temperature is low (around 1100 °C for sample High_SD, see Figure 2), whereas for lower SD systems (i.e., sample Low_SD), the transition temperature is greater (not revealed in our DTA-TG runs for sample Low_SD, but surely lying between 1300 °C and 1500 °C, even though in the DTA-TG run the transition could appear at a temperature higher than 1500 °C, being a dynamic analysis).

Figure 6 shows the plot of Low_SD, Int_SD, and High_SD lattice parameter *a*, obtained by Unit Cell Refinement, as a function of. First, the *a*’s values in Figure 6 are in the same order of the values in Table 1, i.e., *a*’s of Low_SD samples are lower than *a*’s of Int_SD, and these last ones are lower than *a*’s of High_SD. There is only the exception of samples calcined at 800 °C, but in this case, the poor crystallization occurred at such a temperature could impair the evaluation of the lattice parameters.

The analysis of the lattice parameter of the fluorite-like phase offers interesting information about the systems supporting the previous considerations. In all samples, two different trends of the lattice parameter are recognizable: a first one at low temperature (800–1000 °C), characterized by larger values of the lattice parameter, and a second one at high temperature (1100–1300 °C), characterized by smaller values. Considering that the thermal decomposition of all samples ended at 800 °C, the logical inference of these findings is that the chemical composition of the present fluorite-like phase is different between low and high temperatures (i.e., before and after a certain threshold). In the first range of temperatures (800–1000 °C), there is not a well-defined behavior globally characterized by the very slight increase in *a* as a function of the temperature; this could happen as a consequence of two opposite phenomena: the lattice expansion related to heating and the formation of oxygen vacancies related to increasing doping [39]. On the contrary, in the second range of temperature (1100–1300 °C), there is a well-defined behavior with the progressive increase in *a* as a function of the temperature (Figure 6), pointing out that the lattice thermal expansion is the main phenomenon. These findings suggest that a “massive” dissolution of residual Rare-Earth cations in fluorite lattice occurs at temperatures in the range 1000–1100 °C (this agrees with our previous results reported in [4]), in correspondence of the disappearance of the residual amorphous phase. Thus, the fluorite-like phase present at low temperatures is not the real entropy-stabilized single-phase, as it contains only part of the involved cations. Clearly, the crystallization of fluorite occurs before the crystallization of bixbyite (i.e., the most common crystalline phase for RE_2_O_3_ systems [40], where RE stands for a generic trivalent Rare Earth), very likely due to the greater symmetry of the former, and a small portion of the samples remained amorphous also at relatively high temperature (900–1000 °C). Upon the thermal treatment in HTK16, a progressive dissolution of the residual amorphous phase in the fluorite-like phase occurred, giving rise to a further substitutional solid solution that resulted in the highlighted decrease in lattice parameter for all samples between 1000 °C and 1100 °C (Figure 6). In other words, considering the high reactivity of oxidic ceramic powders synthesized via precipitation [41,42] at low temperatures, the direct dissolution of the residual amorphous phase in the as-prepared systems into the fluorite-like phase seems favored upon the competing crystallization and subsequent dissolution of small amounts of bixbyite-like phases. Of course, further analyses are needed to ascertain this hypothesis better, but that was outside the purpose of this study.

Additionally, the lattice parameter of the fluorite-like single phase of all the prepared samples at each temperature in the range 1100–1300 °C follows the same order in Equation (3), again, in perfect agreement with the analogous trend of the average cationic radius.

Thus, the presence at 1300 °C of a main fluorite-like phase in all samples, containing all cations (in the case of High_SD and Int_SD), or most of them in the case of Low_SD, is confirmed.

## 4. Conclusions

Globally considering the results of the high-temperature diffraction analysis, it can be concluded that in the case of fluorite structured entropy-stabilized oxides, the entropy-driven transition temperature from multiple phases (fluorite and the minoritarian bixbyite) to single-phase (fluorite) depends on its specific composition, namely depending on the standard deviation (SD) of its involved cationic radii. Provided that SD is greater than the threshold value (0.095), the phase evolution is qualitatively the same irrespective of the actual equimolar composition (of course belonging to the studied family of compounds), and as a function of the temperature it is represented by:A + F → A + F + B → F + B → F
where A, F, and B stand for amorphous, fluorite, and bixbyite, respectively. Our findings suggest that the last transformation, i.e., the entropy-driven transition from multiphase systems (F + B) to single-phase systems (F), occurs at a temperature depending on SD; in particular, the higher the SD, the lower such a temperature. Moreover, for the highest SD attainable in the analyzed F-ESO systems (based on Cerium, Zirconium, and Rare Earths), the entropy-driven transition temperature lies in the range of 1000 °C to 1100 °C, confirming a previous result published in [4]. For systems owning SD near the fluorite-like single-phase formation threshold (i.e., like the Low_SD sample of this study), the entropy-driven transition temperature lies between 1300 °C and 1500 °C. Thus, generally speaking, for F-ESOs, the entropy-driven transition temperature from multiple phases to fluorite-like single phase is between 1000 °C and 1500 °C.

As the entropy-driven transition temperature is based on both enthalpic contribution and entropic contribution (as highlighted in Equation (2)) and, being the analyzed systems very similar from a chemical point of view, it could be argued that the enthalpic contribution in Gibbs energy should be rather similar, the large difference observed in their entropy-driven transition temperatures would require significant differences in the entropic contribution, not likely based on the chemical composition.

Therefore, based on the conducted basic structural analysis, we propose that the observed differences in the entropy-driven transition temperature of an ESO should be mainly attributed to the difference in the overall entropy owned by that system. In other words, it is suggested that equimolar rare-earth-based multicomponent systems characterized by greater values of SD of the involved cationic radii incorporate a higher amount of entropy and, consequently, such systems can be stabilized in a fluorite-like single-phase at lower temperatures, based on Equation (2). A possible explanation of this phenomenon is related to the enhancement of vibrational effects and, consequently, vibrational entropy due to larger differences in the involved cationic radii when SD is greater.

## Figures and Tables

**Figure 1 materials-16-02219-f001:**
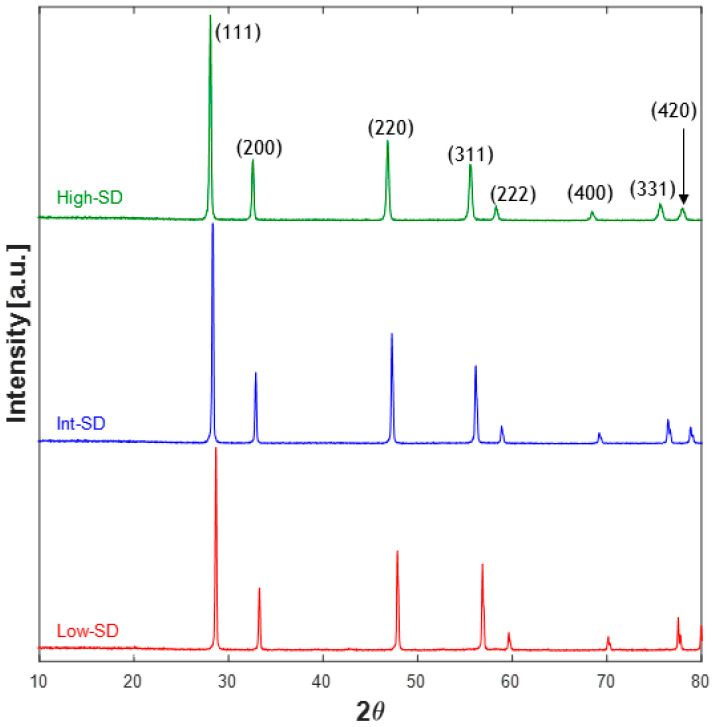
XRD patterns of High_SD, Int_SD, and Low_SD calcined at 1500 °C for 1 h.

**Figure 2 materials-16-02219-f002:**
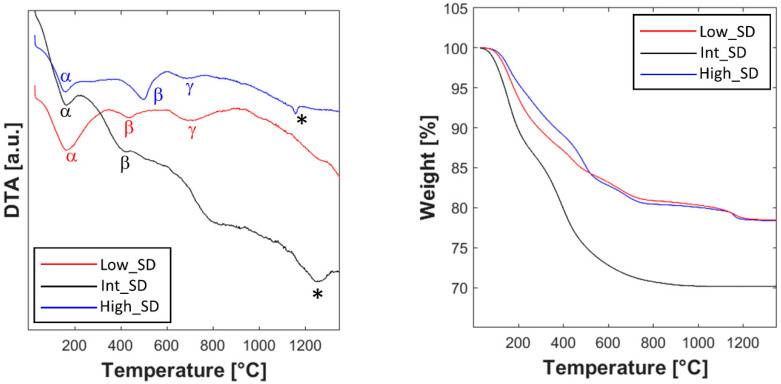
DTA-TG of the as-synthesized samples; * indicates the entropy-driven transition event.

**Figure 3 materials-16-02219-f003:**
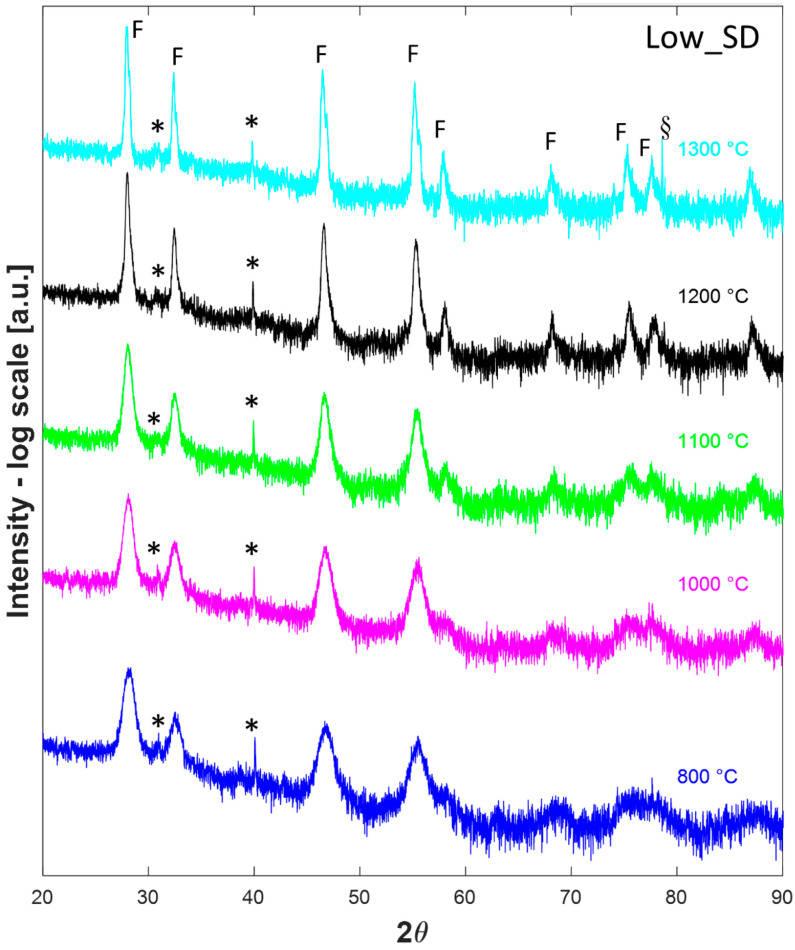
High-temperature (800–1300 °C) diffraction patterns of sample Low_SD. F indicate peaks associated to fluorite, * indicate peaks associated with minoritarian phases, § indicates instrumental artifacts.

**Figure 4 materials-16-02219-f004:**
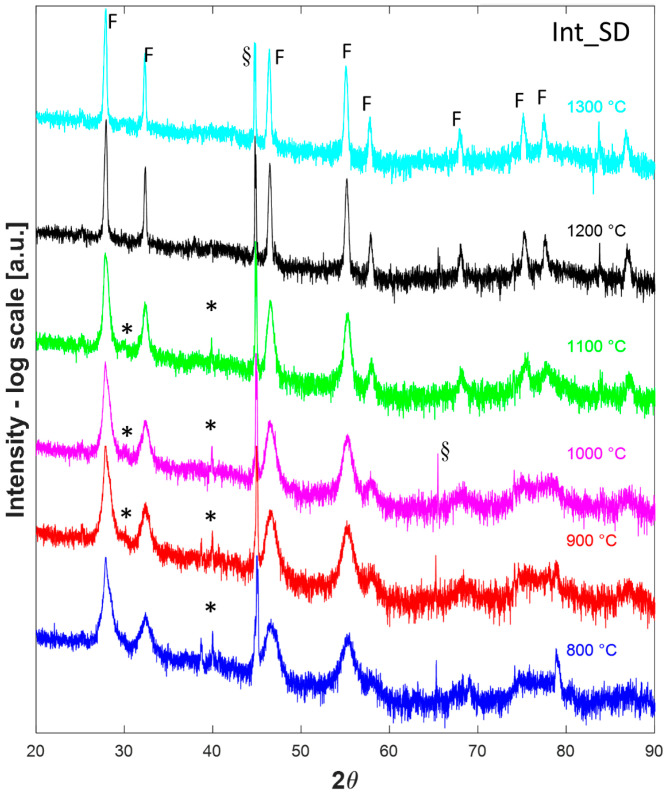
High-temperature (800–1300 °C) diffraction patterns of sample Int_SD. F indicate peaks associated to fluorite, * indicate peaks associated with minoritarian phases, § indicates instrumental artifacts.

**Figure 5 materials-16-02219-f005:**
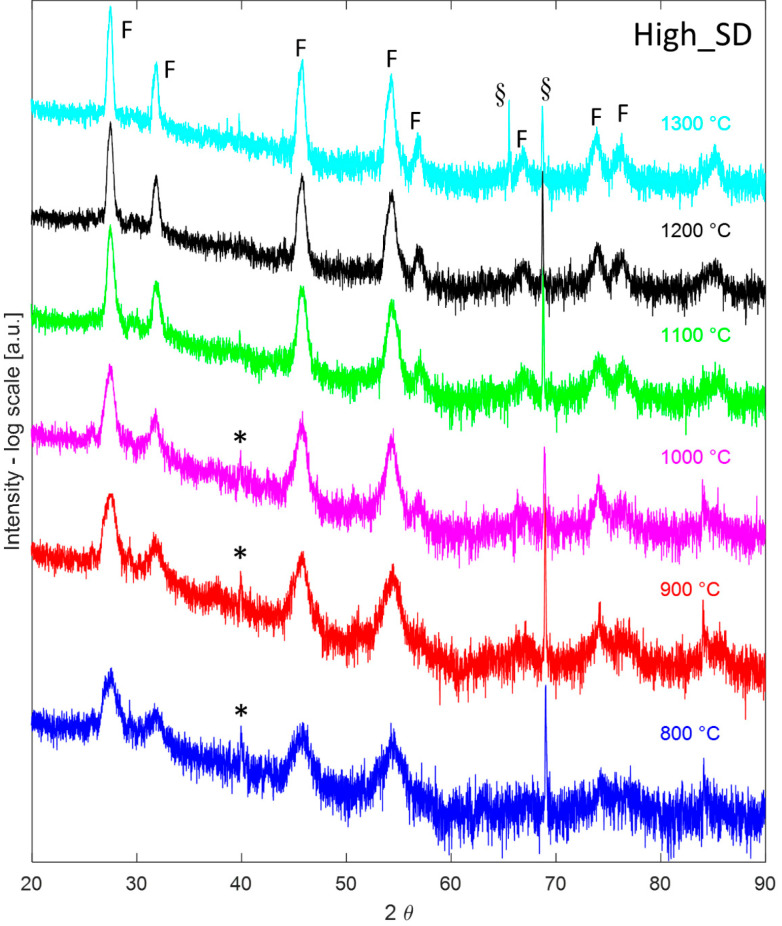
High-temperature (800–1300 °C) diffraction patterns of sample High_SD. F indicate peaks associated to fluorite, * indicate peaks associated with minoritarian phases, § indicates instrumental artifacts.

**Figure 6 materials-16-02219-f006:**
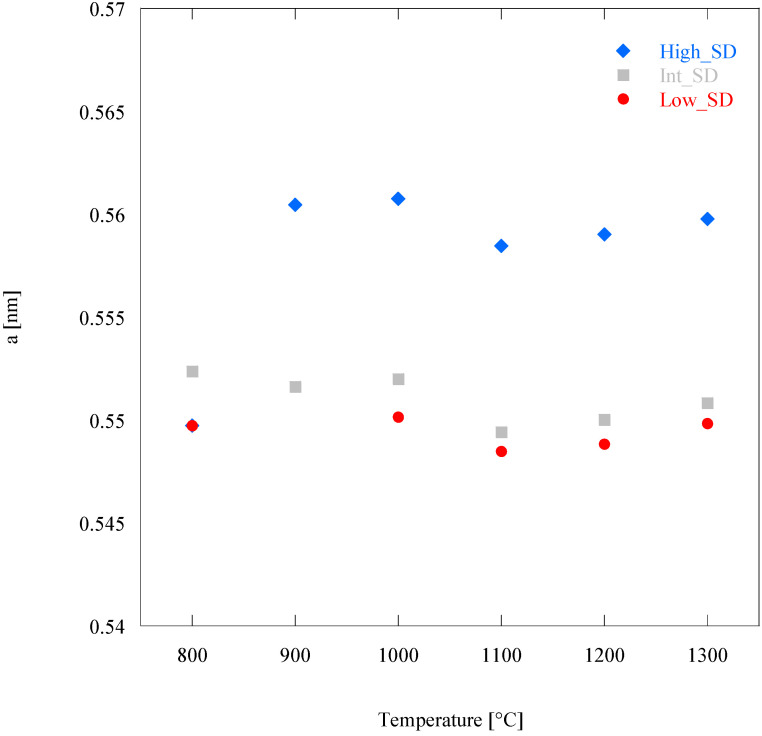
Lattice Parameters of Low_SD, Int_SD and High_SD as a function of temperature.

**Table 1 materials-16-02219-t001:** Composition of samples and data of cationic radius of the present cations.

Chemical Formula	Label	r_Ce^4+^ [Å]	r_Zr^4+^ [Å]	r_RE1^3+^ [Å]	r_RE2^3+^ [Å]	r_RE3^3+^ [Å]	SD [Å]	$\bar{r}$ [Å]
Ce_0.2_Zr_0.2_Nd_0.2_Y_0.2_Er_0.2_O_1.7_	Low_SD	0.97	0.84	1.109	1.019	1.004	0.0976	0.988
Ce_0.2_Zr_0.2_La_0.2_Y_0.2_Gd_0.2_O_1.7_	Int_SD	0.97	0.84	1.16	1.019	1.053	0.117	1.008
Ce_0.2_Zr_0.2_La_0.2_Nd_0.2_Sm_0.2_O_1.7_	High_SD	0.97	0.84	1.16	1.109	1.079	0.128	1.031

**Table 2 materials-16-02219-t002:** Lattice parameter and crystal size of samples calcined at 1500 °C for 1 h.

Sample	$\bar{r}$ [nm]	*a* (nm)	Crystal Size [nm]
Low_SD	0.0988	0.536031 (1)	86.9 (2)
Int_SD	0.1008	0.542480 (6)	76.9 (5)
High_SD	0.1031	0.547973 (1)	49.5 (4)

## Data Availability

The data presented in this study are available on request from the corresponding author. The data are not publicly available because they are properties of different Academic subjects.

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
