# Peer review of "On the Effect of Standard Deviation of Cationic Radii on the Transition Temperature in Fluorite-Structured Entropy-Stabilized Oxides (F-ESO)"

_materials, 2023, doi:10.3390/ma16062219_

Round 1

Reviewer 1 Report

Interesting results are obtained in the fluorite-structured entropy-stabilized oxides. The work can be published after a minor revision as follows.

1. The symbols in Line 170, 171 cannot be recognized in the manuscript.

2. The lattice parameter a is largest in High_SD then Int_SD and Low_SD. However, it shows the opposite trend in Fig. 6 as reflected by the high-Temperature XRD results. The authors should further explain how the second phase or the amorphous phase affects the grain size and lattice parameter during the process of the single-phase formation. The paper (DOI: 10.1149/2162-8777/acb28e) may be useful and could be cited.

3. The SEM are suggested, and it would be better if the DTA up to 1500 oC for Low_SD is provided (not neccesary).

4. In Fig. 6c, the right blue dot should correspond to 1300 oC?

5. What is the relationship about the SD and the single-phase formation in fluorite-structured high-entropy oxides?

Author Response

Replies to Reviewer 1

Interesting results are obtained in the fluorite-structured entropy-stabilized oxides. The work can be published after a minor revision as follows.

  1. The symbols in Line 170, 171 cannot be recognized in the manuscript.

We thank the reviewer for his/her comment. Symbols in lines 170 and 171 have been correctly added (being a, b, and g, respectively).

  1. The lattice parameter a is largest in High_SD then Int_SD and Low_SD. However, it shows the opposite trend in Fig. 6 as reflected by the high-Temperature XRD results. The authors should further explain how the second phase or the amorphous phase affects the grain size and lattice parameter during the process of the single-phase formation. The paper (DOI: 10.1149/2162-8777/acb28e) may be useful and could be cited.

We thank the reviewer for his/her suggestion, as we needed to be clearer in the description of Figure 6. Actually, all the values of “a” reported in Figure 6 are in the same order as the corresponding values reported in Table 2, with the only exception of data at 800°C at which the values are all very similar to each other (we believe that it depends on the poor crystallization occurred at such temperature). We have slightly reworded the text in the manuscript to clarify that better. Concerning the second point raised by the reviewer, we thank him/her for the suggested paper (also cited in the bibliography as ref [39]), as it has been very useful to us. Generally, the presence of a second or amorphous phase may influence the grain size and the lattice parameter. In the present study, we believe that the presence of the amorphous phase, as a “source” of rare-earth dopants during the calcination step, may influence the process. We have extended and reworded the discussion of Figure 6 in the text.

The SEM are suggested, and it would be better if the DTA up to 1500 °C for Low_SD is provided (not necessary).

We agree with the reviewer, as the SEM could help provide additional data to highlight morphological differences related to different values of SD. Still, unfortunately, at the moment, the SEM in our laboratory is out-of-service, and we cannot reach another one soon, considering the time for the revision. Anyway, the main focus of the study was to analyze the effect of SD on the transition temperature. As the transition temperature lies only on thermodynamic parameters, morphological differences, despite their intrinsic values, are not the main point of our experimental work. For that, we did not access other laboratories to conduct SEM observations.

  1. In Fig. 6c, the right blue dot should correspond to 1300 °C?

The reviewer is right: it was a typo. A new Figure 6 has been added within the revised manuscript, correctly reporting the data at 1300 °C.

  1. What is the relationship about the SD and the single-phase formation in fluorite-structured high-entropy oxides?

In our previous paper (L. Spiridigliozzi, C. Ferone, R. Cioffi, G. Dell'Agli, A simple and effective predictor to design novel fluorite-structured High Entropy Oxides (HEOs), Acta Materialia, 202 (2021) 181-189), we explained such a relationship. Specifically, we showed that considering rare earth oxides (including zirconium oxide), the formation of a single fluorite-like phase is possible via thermal treatment at high temperatures if the SD of cationic radii is greater than a threshold value (s = 0.095). Conversely, in this work, we have shown that the temperature necessary to induce the formation of a single entropy-stabilized phase depends on the SD, too (which should be larger than the threshold value). In other words, the SD of cationic radii of a certain rare-earth-based entropy-stabilized oxide not only acts as a predictor of the formed phase but also influences its entropy-driven transition temperature from a multiphase system to a single-phase one.

Reviewer 2 Report

Dear Authors,

Thank you for your work.

Article entitled On the effect of standard deviation of cationic radii on the transition temperature in Fluorite-structured Entropy-Stabilized Oxides (F-ESO) is thematically related to the subject of the journal. The paper contains interesting content from a scientific point of view, but the research and literature analysis has not been sufficiently disclosed.

The Authors describe phase transfer, phase crystallization, they use the term "other phases", but it is difficult to find information on what, apart from temperature, affects the quality, crystallization or potential number of phases. Could the Authors introduce information on amorphous phases next to the topic of crystallization? I do not consider the title to be fully consistent with the content of the work - can you refine the introduction and analysis of the literature and the research scheme to the topic or topic to the presented research.

The article describes the results of the experiment, but there is no more information about the research process, I would like to ask for supplementation, because point 2. Experimental is contained on 1 page, and then the Authors describe the Results and the Discussion. I also suggest separating these 2 points and specifying exactly what is the result and what is debatable? The parameters of the used test equipment/machines and the database for the XRD test should also be taken into account.

Other comments below.

1) The Authors wrote:

"As reported in Table 1, all the analyzed samples own a SD of the involved cationic radii greater than the threshold value reported in our previous work [23], approximately equal to 0.095" - I believe that if this is a continuation of research, the most important information should be included in this article. I suggest the authors take into account the foundations of the previous article and what distinguishes the previous article from the current one - what is the research progress, it should be clearly specified.

2) It happens that some phases occur below the angle of 20 during the XDR test, I understand that in this case there was no such possibility?

3) Little information is given about the possible directions of crystallization. 1 sentence on this topic appeared in line 268. Can the Authors expand on this topic?

4) The conclusions again mention the phases in general, and additionally that the phases depend on the composition - of course this is a correct statement, and going back to the comment above, please describe what influences (apart from the discussed temperature) the crystallization and synthesis of the phases? Please elaborate on the comment: it can be concluded that the entropy-driven transition temperature from multiple phases to single phase of an equimolar ESO depends on its specific composition, namely depending on the standard deviation (SD) of its involved cationic radii.”

Regards,

Reviewer

Author Response

Replies to Reviewer 2

Article entitled On the effect of standard deviation of cationic radii on the transition temperature in Fluorite-structured Entropy-Stabilized Oxides (F-ESO) is thematically related to the subject of the journal. The paper contains interesting content from a scientific point of view, but the research and literature analysis has not been sufficiently disclosed.

The Authors describe phase transfer, phase crystallization, they use the term "other phases", but it is difficult to find information on what, apart from temperature, affects the quality, crystallization or potential number of phases. Could the Authors introduce information on amorphous phases next to the topic of crystallization? I do not consider the title to be fully consistent with the content of the work - can you refine the introduction and analysis of the literature and the research scheme to the topic or topic to the presented research.ert

The reviewer is right, as we were unclear on the points raised by him/her. In the studied systems, we can have one amorphous phase and two different crystalline phases during the process, i.e., fluorite phase (main phase) and bixbyite (very minoritarian phase), confirming the findings of our previous studies (see also replies to Reviewer n.2). During the calcination runs, at temperatures greater than 800°C, the residual amorphous phase gradually dissolved in the fluorite-like phase and eventually also the minoritarian bixbyite-like phase dissolve in the fluorite lattice (this last transformation is the transition to entropy stabilized oxide and occurs at a temperature variable with the SD). According also to Reviewer n.2, we have markedly reworded the discussion of the results of the HTK analysis (Figures 3, 4, and 5) to clarify our analysis better.

Actually, we do not have detailed information on the amorphous phase formed during the coprecipitation process (the as-synthesized products are formed by ill-crystallized fluorite phase, formed by CeO2 possibly with a certain level of doping, and by a major amorphous phase, as shown in Figure S1). During the thermal treatment step, the amorphous phase crystallized. Still, there are neither sharp nor broad exothermic peaks in the thermographs of all three samples suggesting that the crystallization occurred gradually and in a wide range of temperatures. This behavior is related to the chemical complexity of the studied systems; in systems based on only ceria or single rare-earth doped-ceria, there is a broad but evident crystallization peak at low temperature (L. Spiridigliozzi, G. Dell’Agli, M. Biesuz, V.M. Sglavo, M. Pansini, “Effect of the Precipitating Agent on the Synthesis and Sintering Behaviour of 20 mol% Sm-doped Ceria”, Advances in Materials Science and Engineering, Volume 2016, Article ID 6096123, 8 pages, doi:10.1155/2016/6096123) indicating a well-defined crystallization event. We have added this point to the manuscript. Finally, a detailed analysis of the amorphous phase was outside the purpose of the present work. Concerning the title, we prefer to keep the current title because it recalls the main focus of the work, which is the relation between the SD and the multiphase to single-phase entropy-driven transition temperature in the fluorite-structured entropy-stabilized oxides.

The article describes the results of the experiment, but there is no more information about the research process, I would like to ask for supplementation, because point 2.

We thank the reviewer for his/her suggestion. We have added some further details concerning the process in the experimental part.

Experimental is contained on 1 page, and then the Authors describe the Results and the Discussion. I also suggest separating these 2 points and specifying exactly what is the result and what is debatable? The parameters of the used test equipment/machines and the database for the XRD test should also be taken into account.

We thank the reviewer for his/her suggestion. Accordingly, we have added the instrumental parameter for diffraction analysis; in the case of DTA-TG runs, the relevant information was already given in the manuscript. We have added the reference card of the ICDD database. Concerning the separation of the section “Results and Discussion”, we are not able to meet this suggestion as the manuscript had been structured with only a section for the sake of reading continuity.

Other comments below.

  • The Authors wrote: "As reported in Table 1, all the analyzed samples own a SD of the involved cationic radii greater than the threshold value reported in our previous work [23], approximately equal to 0.095" - I believe that if this is a continuation of research, the most important information should be included in this article. I suggest the authors take into account the foundations of the previous article and what distinguishes the previous article from the current one – what is the research progress, it should be clearly specified.

As correctly stated by the reviewer, this work logically continues from our previous work. However, there is a major difference between the two works; as in the already published one, we demonstrated that SD of the cationic radii of a certain rare-earths-based multi-cationic system determines the entropy-driven stabilization of either a bixbyite-like or a fluorite-like crystal phase; conversely, in the present manuscript, we demonstrated that SD also has a direct effect on the transition temperature for a certain rare-earths-based multi-cationic system. In other words, higher SDs are directly related to the entropic contribution of such multi-cationic systems.

  • It happens that some phases occur below the angle of 20 during the XRD test, I understand that in this case there was no such possibility?

We thank the reviewer for his/her correct observation. However, we started the XRD acquisitions from 20°, and our systems’ possible crystalline phases (see also our reply to point 4) do not exhibit reflections at angles lower than 20° (see also Figure 1 in “L. Spiridigliozzi, C. Ferone, R. Cioffi, G. Dell'Agli, A simple and effective predictor to design novel fluorite-structured High Entropy Oxides (HEOs), Acta Materialia, 202 (2021) 181-189”).

  • Little information is given about the possible directions of crystallization. 1 sentence on this topic appeared in line 268. Can the Authors expand on this topic?

We thank the reviewer for his/her observation. However, in line 268 of the manuscript version visible to us, there are not any comments concerning crystallization; actually, it is discussed the effect of the temperature on the fluorite lattice parameter. Anyway, regarding possible directions for the crystallization of the as-synthesized coprecipitated products, we have already replied to the reviewer’s first comment. This point has been expended both in the discussion and in the conclusion of the manuscript.

  • The conclusions again mention the phases in general, and additionally that the phases depend on the composition - of course this is a correct statement, and going back to the comment above, please describe what influences (apart from the discussed temperature) the crystallization and synthesis of the phases? Please elaborate on the comment: it can be concluded that the entropy-driven transition temperature from multiple phases to single phase of an equimolar ESO depends on its specific composition, namely depending on the standard deviation (SD) of its involved cationic radii.”

A detailed discussion concerning the oxide-based phases that can be synthesized and crystallized in the studied compositional space (five oxides selected among rare-earths and zirconium with equimolar composition, and Ce always present) was reported in our previous work. Summarizing these results, the possible equilibrium crystalline phases in this compositional space may be bixbyite, fluorite, or a mixture of them; in this work, we have studied three particular compositions for which our predictor based on SD predicts fluorite as the equilibrium phase. On the contrary, the as-synthesized products are mainly amorphous in nature with the presence of an ill-crystallized fluorite phase, deriving them from a coprecipitation process; during the heating, the crystallization process leads firstly to multiphase (fluorite plus bixbyite) and beyond the transition temperature only fluorite. The diffraction data collected by HTK16 confirmed that. We have also expanded and commented in the conclusion section on the sentence indicated by the reviewer.

Reviewer 3 Report

In the manuscript entitled “On the effect of standard deviation of cationic radii on the transition temperature in Fluorite-structured Entropy-Stabilized Oxides (F-ESO)”, authors develop three F-ESO compositions with different value of standard deviation synthesized via co-precipitation and characterized by X-ray diffraction as a function of temperature and thermal analysis (DSC and TGA).

Although the manuscript could present some interesting results for the development of new high entropy oxide systems, the manuscript, in the present form, lacks sufficient clearness to be recommended to publish in Materials for several reasons:

·         Typos. Some examples. Line 170, 121, 120…

·         A more detailed discussion of prior literature in this area and a clear discussion of how results presented here are truly novel. Some important information is missing. For example, there are two widely accepted ways of defining high entropy systems, based on composition and entropy. Moreover, there are in literature examples of spinel high entropy oxides which should be appropriately referred. HEOs systems have also been reported as interesting materials for its magnetic properties.

·         The election of the chosen composition should be justified.

·         Table 1. It should be specified how SD is calculated and the precedence of the data of cationic radius. Please, include errors.

·         Authors should explain the air quenching performed.

·         Authors should clarify how the x-ray diffraction experiments have been performed. It is not clear. It should be specified the heating rate and the atmosphere.

·         Table 2. Please, include estimated errors. It would be interesting to carry out a fitting of the XRD patterns (LeBail or Rietvield) in order to obtain more accurate values.

·         Regarding X-ray diffraction pattern, some points remains unclear for me. Authors claim that the phase transformation occurs at temperature higher 1100ºC. However, in the XRD patterns as a function of temperature (Figs. 3-5) in all the patterns the same diffraction patterns can be observed. It looks like just an increase of the crystal size. It should be clarified.

·         Figure 6. The different panel should be unified in order to facilitate the comparison between the different compounds

Author Response

Replies to Reviewer 3

In the manuscript entitled “On the effect of the standard deviation of cationic radii on the transition temperature in Fluorite-structured Entropy-Stabilized Oxides (F-ESO),” authors develop three F-ESO compositions with different values of standard deviation synthesized via co-precipitation and characterized by X-ray diffraction as a function of temperature and thermal analysis (DSC and TGA). Although the manuscript could present some interesting results for the development of new high entropy oxide systems, the manuscript, in the present form, lacks sufficient clearness to be recommended for publish in Materials for several reasons:

  • Some examples. Line 170, 121, 120…

We thank the reviewer for his observation. We’ve checked the manuscript and corrected several typos within the text.

  • A more detailed discussion of prior literature in this area and a clear discussion of how the results presented here are genuinely novel. Some critical information is missing. For example, there are two widely accepted ways of defining high entropy systems, based on composition and entropy. Moreover, there are in literature examples of spinel high entropy oxides which should be appropriately referred. HEOs systems have also been reported as interesting materials for its magnetic properties.

We thank the reviewer for his/her suggestion. Accordingly, we have extended the manuscript's introduction to meet the reviewers' requirements, and we have also modified several points of the discussion to clarify our results. The novelty of the work is also better highlighted in the Conclusion.

  • The election of the chosen composition should be justified.

We thank the reviewer for his/her suggestion, as we needed clarification on this critical point. The compositions of the three investigated systems were selected to have all the SD greater than 0.095 (the threshold value reported in our previous work needed to obtain an entropy-stabilized fluorite-like phase); given that, the first sample has been designed to have a SD only slightly greater than 0.095 (namely 0.0976), another one has been designed to have a very high SD (namely 0.128). In contrast, the last one has been designed to have an intermediate SD value between the other two (namely 0.117). We have clarified this better in the introduction.

  • Table 1. It should be specified how SD is calculated and the precedence of the data of cationic radius. Please, include errors.

The equation for the calculus of SD (the same one also reported in our previous work) has been added in the Introduction. We assumed the exact values of the cationic radii from “Shannon, R. D. (1976). Revised effective ionic radii and systematic studies of interatomic distances in halides and chalcogenides. Acta Crystallographica Section A: crystal physics, diffraction, theoretical and general crystallography, 32(5), 751-767”, so that no error could be calculated for the SD. We added such detail within the revised manuscript.

  • Authors should explain the air quenching performed.

The air quenching was carried out this way: the sample was drawn from the furnaces at 1250°C and cooled in the air; the cooling rate was high at the beginning, and of course, it slowed with the decrease of the temperature. After a few minutes, the platinum crucible could be manipulated by hand.

  • Authors should clarify how the x-ray diffraction experiments have been performed. It is not clear. It should be specified the heating rate and the atmosphere.

We thank the reviewer for his/her suggestion. More details have been added in the Experimental section about the in-situ XRD experiments, describing, in particular, the thermal cycling and the operation in air atmosphere.

  • Table 2. Please, include estimated errors. It would be interesting to carry out a fitting of the XRD patterns (LeBail or Rietvield) in order to obtain more accurate values.

According to the reviewer’s advice, we have carried out Rietveld refinement of the three diffraction patterns of samples calcined at 1500°C. The relative plots are reported in the Supplementary Materials, exhibiting a very good fit. Table 2 was updated with the new values of a and of crystal size, including the estimated errors reported as standard deviations on the last significant digit.

  • Regarding X-ray diffraction pattern, some points remains unclear for me. Authors claim that the phase transformation occurs at temperature higher 1100ºC. However, in the XRD patterns as a function of temperature (Figs. 3-5) in all the patterns the same diffraction patterns can be observed. It looks like just an increase of the crystal size. It should be clarified.

We thank the reviewer for his/her comment, as we may have needed to clarify this point better in discussing the results. Although the diffraction patterns in Figures 3-5 appear similar, some minor differences are representative of the presence of either a single-phase system or a multiphase one. In these Figures, the secondary phase (very likely bixbyite, as already obtained in similar systems) is represented by stars “*” (at angles slightly greater than 30° 2q and about 40° 2q); so by increasing the temperature in some samples, those peaks disappeared, indicating the disappearance of the secondary phase leaving a single fluorite phase, i.e., the transition temperature had been overcome (it occurs in samples Int_SD, Figure 4, and High_SD, Figure 5). However, in the sample Low_SD that did not happen because the transition temperature was in the range of 1300-1500 °C. The presence of such weak peaks associated with the bixbyite-like phase was also revealed in our previous study [Ref 4 and 27]. Their appearance and disappearance as a function of the temperature are typical of the entropy-driven stabilization. For the sake of clarity, we reworded this point within the revised manuscript.

  • Figure 6. The different panel should be unified in order to facilitate the comparison between the different compounds.

We thank the reviewer for his/her suggestion. We updated Figure 6 accordingly.

Reviewer 4 Report

The authors have presented an interesting study intro entropy stabilisation of single phase assemblage in oxide materials. The results and their discussion pose some good information that will be useful to many others doing research in this and related field. In general the field of high entropy oxides is rapidly growing in its size. In particular the relationship between formation and cation size distribution is thought provoking.

Thank you for your contribution to the field

Author Response

Replies to Reviewer 4

The authors have presented an interesting study intro entropy stabilisation of single phase assemblage in oxide materials. The results and their discussion pose some good information that will be useful to many others doing research in this and related field. In general the field of high entropy oxides is rapidly growing in its size. In particular the relationship between formation and cation size distribution is thought provoking.

Thank you for your contribution to the field

We thank the reviewer for his/her appreciation of our work. We hope to continue contributing with additional results in the field of High-Entropy Materials.

Round 2

Reviewer 2 Report

Good morning,

The Authors write about the standard deviation, focusing mainly on the XRD study, and then DTA would not be enough. That is why I think that the topic should contain the statement: 'based on the basic structural analysis', which is suggested by the XRD study, and then DTA (although this is very basic research).

Amorphic phases are described, exactly 'the disappearance of residual amorphous phases', are the Authors able to write anything about these type of phases?

These phases are crystallized as a result of, among others temperature operation.

Gibbs energy is described, but where the information is about it - I refer to the page e.g.: https://thermoddem.bgm.fr/

Author Response

Replies to Reviewer 2

The Authors write about the standard deviation, focusing mainly on the XRD study, and then DTA would not be enough. That is why I think that the topic should contain the statement: 'based on the basic structural analysis', which is suggested by the XRD study, and then DTA (although this is very basic research).

With this work, we intended to reveal the possible effect of SD of the cationic radii in 5-components rare-earth-based ESOs on the transition temperature from a multiphase system (fluorite + bixbyite) to a single-phase system (fluorite) when the SD is greater than 0.095, so that fluorite is the single phase stable at high temperatures. The main way to demonstrate such phase transformations is the use of X-Ray diffraction. Therefore, our work was mainly based on powder diffraction data.  Anyway, we carried out DTA-TG analysis on the various samples, too, revealing weak endothermic signals (for two samples) attributable to the transformation B + F à F, confirming the XRD-driven results.

For the sake of clarity, we have added the statement suggested by the reviewer both in the abstract and within the main text (Introduction and Conclusions sections).

Amorphic phases are described, exactly 'the disappearance of residual amorphous phases', are the Authors able to write anything about these type of phases?

We were unclear on the point raised by the reviewer. Very likely, the amorphous phase present in the as-synthesized products at room temperature is a mixture of hydroxides and hydrated oxides (as obtained for ceria/zirconia-based systems in our previous works: “Spiridigliozzi, L., Dell’Agli, G., Biesuz, M., Sglavo, V. M., & Pansini, M. (2016). Effect of the precipitating agent on the synthesis and sintering behavior of 20 mol Sm-doped ceria. Advances in Materials Science and Engineering” and “Spiridigliozzi, L., Ferone, C., Cioffi, R., Accardo, G., Frattini, D., & Dell’Agli, G. (2020). Entropy-stabilized oxides owning fluorite structure obtained by hydrothermal treatment. Materials, 13(3), 558”), losing water upon heating and leaving residual amorphous phase formed by oxides at higher temperatures (greater than 800°C) in addition to crystallized products. We did not study in detail the specific features of the amorphous phase present in the various samples, because our work was focused on the identification of the crystalline phases present at high temperatures.

These phases are crystallized as a result of, among others temperature operation.

The reviewer is right, as the increasing temperature influences the crystallization of the amorphous phase. However, in the case of non-entropy-stabilized-oxides, the amorphous precursors would have been crystallized, upon the increasing temperature, as single oxides owning different crystal structures.

Gibbs energy is described, but where the information is about it - I refer to the page e.g.: https://thermoddem.bgm.fr/

Actually, information about the exact value of the parameter present in the expression of Gibbs energy is not available for our complex systems. Within the manuscript, Equation (2) only shows the terms necessary to evaluate G, in order to highlight what influences it and, in particular, the parameters influencing the entropy-driven transition temperature.

Reviewer 3 Report

I appreciate the effort of authors in order to improve the quality of the manuscript. I recommend accept it in present form

Author Response

Replies to reviewer 3

I appreciate the effort of authors in order to improve the quality of the manuscript. I recommend accept it in present form.

We thank the reviewer for his/her positive comment. We are glad we improved our manuscript thanks to all reviewers’comments.
